# Longitudinal assessment of DREADD expression and efficacy in the monkey brain

Yuji Nagai[1]*, Yukiko Hori[1], Ken-ichi Inoue[2], Toshiyuki Hirabayashi[1], Koki Mimura[1], Kei Oyama[1], Naohisa Miyakawa[1], Yuki Hori[1], Haruhiko Iwaoki[1], Katsushi Kumata[3], Ming-Rong Zhang[3], Masahiko Takada[1], Makoto Higuchi[1], Takafumi Minamimoto[1]*

[1]Advanced Neuroimaging Center, National Institutes for Quantum Science and Technology, Chiba, Japan; [2]System Neuroscience Section, Center for the Evolutionary Origins of Human Behavior, Kyoto University, Inuyama, Japan; [3]Department of Advanced Nuclear Medicine Sciences, National Institutes for Quantum Science and Technology, Chiba, Japan

*For correspondence:
nagai.yuji@qst.go.jp (YN);
minamimoto.takafumi@qst.go.
jp (TM)

Competing interest: The authors declare that no competing interests exist.

## eLife Assessment

This study provides novel and **fundamental** insights into the long-term use of DREADDs to modulate neuronal activity in nonhuman primates. The **exceptional** evidence demonstrates the peak dynamics and the subsequent stability of chemogenetic effects for 1.5 years, informing the experimental designs and the interpretation of highly impactful chemogenetic studies in macaques. The protocols, data, and outcomes can serve as guidelines for future experiments. Therefore, the findings will be of significant interest to the field of chemogenetics and may also be of broader interest to researchers and clinicians who seek to utilize viral vectors and/or related genetic technologies.

**Abstract** Designer Receptors Exclusively Activated by Designer Drugs (DREADDs) offer a powerful means for reversible control of neuronal activity through systemic administration of inert actuators. Because chemogenetic control relies on DREADD expression levels, understanding and quantifying the temporal dynamics of their expression is crucial for planning long-term experiments in monkeys. In this study, we longitudinally quantified in vivo DREADD expression in macaque monkeys using positron emission tomography with the DREADD-selective tracer [11C]deschloroclozapine (DCZ), complemented by functional studies. Twenty macaque monkeys were evaluated after being injected with adeno-associated virus vectors expressing the DREADDs hM4Di or hM3Dq, whose expression was quantified as changes in [11C]DCZ binding potential from baseline levels. Expression levels of both hM4Di and hM3Dq peaked around 60 days post-injection, remained stable for about 1.5 years, and declined gradually after 2 years. Significant chemogenetic control of neural activity and behavior persisted for about 2 years. The presence of protein tags significantly influenced expression levels, with co-expressed protein tags reducing overall expression levels. These findings provide valuable insights and guidelines for optimizing the use of DREADDs in long-term primate studies and potential therapeutic applications.

## Introduction

Chemogenetic technology affords remote and reversible control of neuronal activity by expressing receptors that are designed to be activated by systemically delivered biologically inert actuators. One such tool is the widely used Designer Receptors Exclusively Activated by Designer Drugs (DREADD)

system, which is based on mutated muscarinic receptors (*Roth, 2016*). DREADDs have been widely used in rodent models and their use in nonhuman primates has recently shown some promise by inducing significant behavioral changes through simultaneous modulation of activity in disparate neuronal populations across large brain regions (*Nagai et al., 2016*; *Nagai et al., 2016*). Whether this promise will translate to reality depends on how long DREADDs can exert their effects. Because the efficacy of chemogenetic manipulation depends on DREADD expression levels (*Grayson et al., 2016*; *Upright et al., 2018*), maintaining high and stable expression levels is critical for ensuring consistent control throughout all experiments in a study. While this has proven possible in rodent studies that typically only span several months, whether it is true for studies involving macaque monkeys, which often last multiple years, remains a question. Indeed, despite the growing use of DREADDs in monkey neuroscience research, consistent and reliable data on the temporal dynamics of DREADD expression following viral transduction remain scarce. Specifically, little is known about how quickly DREADD expression is established following viral transduction or how long it is maintained. This gap in knowledge complicates efforts to optimize critical parameters, such as the serotypes, constructs, and titers of viral vectors that all influence expression level and duration.

Positron emission tomography (PET) is an in vivo imaging technique that allows quantification of receptor density in the brain. Several studies have demonstrated the utility of PET for longitudinal measurements of gene expression introduced into the brain via adeno-associated virus (AAV) (*Nerella et al., 2023*; *Vandeputte et al., 2011*; *Yoon et al., 2014*). While these studies have highlighted the potential of using PET imaging to monitor gene expression over time, most have used it as an indirect measure of target gene expression, without exploring the relationship between expression levels and the resultant functional outcomes.

Several PET probes have been developed to visualize DREADD receptor expression. Initial studies in rodents used [$^{11}$C]clozapine-N-oxide (CNO) and [$^{11}$C]clozapine to monitor DREADD expression in the brain (*Ji et al., 2016*). In nonhuman primates, longitudinal PET studies using [$^{11}$C]clozapine have successfully tracked virally expressed DREADDs in macaques (*Nagai et al., 2016*). More recently, DREADD-selective ligands such as [$^{18}$F]JHU37107 (*Bonaventura et al., 2019*) and [$^{11}$C]deschlorocclozapine ([$^{11}$C]DCZ) *Nagai et al., 2020* have been developed. Among these, [$^{11}$C]DCZ has become a particularly useful tracer for studying DREADD-mediated functional manipulation in macaques and marmosets. This PET ligand allows researchers to verify the location and density of DREADD expression at the site of AAV injection and at projection sites, as demonstrated in numerous studies (*Hirabayashi et al., 2024*; *Hirabayashi et al., 2021*; *Hori et al., 2021*; *Mimura et al., 2021*; *Miyakawa et al., 2023*; *Miyake et al., 2024*; *Mueller et al., 2023*; *Nagai et al., 2020*; *Nagai et al., 2016*; *Oyama et al., 2024*; *Oyama et al., 2022a*; *Oyama et al., 2021*; *Roseboom et al., 2021*). Moreover, unlike postmortem analysis that relies on the detection of fused or expressed protein tags, [$^{11}$C]DCZ PET provides a direct, quantitative, and noninvasive measure of DREADD expression level in vivo, using standard reference tissue models (*Nagai et al., 2020*; *Yan et al., 2021*). [$^{11}$C]DCZ PET thus serves as a valuable tool for tracking DREADD expression levels throughout long-term chemogenetics experiments in monkeys.

As the developer of the DREADD-selective agonist DCZ (*Nagai et al., 2020*), we have consistently performed multiple [$^{11}$C]DCZ PET scans as part of our DREADD experiments in monkeys, including at baseline before AAV injection, 30–120 days post-injection, and then periodically over the course of the experiments. For the current study, we analyzed these PET data to examine the short- and long-term dynamics of DREADD expression in vivo and assessed how these changes correlated with the ability to manipulate neuronal activity or behavior. Our results provide valuable information with a few caveats and indicate that the promise of using DREADDs in long-term monkey experiments is real.

## Results

PET imaging data using [$^{11}$C]DCZ were collected from 15 monkeys before (baseline) and after injections of AAV vectors for expressing hM4Di or hM3Dq (see *Table 1*). To achieve neuron-specific expression, we used AAV vectors with a preference for neuronal infection, such as AAV2 or AAV2.1 (*Kimura et al., 2023*), or for incorporation with neuron-specific promoters (e.g., human synapsin promoter; hSyn). To assess the specific binding of [$^{11}$C]DCZ, we calculated the binding potential relative to a nondisplaceable radioligand (BP$_{ND}$) using a multilinear reference tissue model with the cerebellum as a reference region (*Yan et al., 2021*). DREADD expression levels were then quantified as the change

in BP$_{ND}$ (ΔBP$_{ND}$) from baseline values. This approach allowed us to longitudinally and quantitatively monitor the dynamics of DREADD expression in vivo.

## Peak DREADD expression occurred approximately 60 days post-injection

To evaluate the temporal profile of DREADD expression and determine how quickly expression was established following transduction, we analyzed the [$^{11}$C]DCZ PET data from seven monkeys that received AAV vector injections into subcortical regions with similar volumes (*Tables 1 and 2*). These data were originally collected for vector-testing studies, during which PET scans were frequently repeated up to 150 days post-injection. The dataset was selected according to the following criteria: (1) the titer of injected viral vector was 1.0–3.0×10$^{13}$ gc/ml, and (2) the peak ΔBP$_{ND}$ value exceeded 0.5. Analysis showed that expression levels of hM4Di and hM3Dq increased rapidly, reaching peak expression at approximately 60 days after viral vector injection. Notably, no clear differences in expression dynamics were observed between hM3Dq and hM4Di (*Figure 1*).

**Table 1.** Summary of subjects, DREADD used, and functional assessments used in this study.

| ID | Species | Sex | weight (kg) | Age (years) | DREADD | [$^{11}$C]DCZ | Behavior | FDG | Electrophysiology |
|---|---|---|---|---|---|---|---|---|---|
| 153 | R | M | 6.8 | 10 | hM4Di | | ✓* | | |
| 163 | R | M | 6.1 | 13 | hM4Di | | ✓† | | |
| 193 | R | M | 4.4 | 9 | hM4Di | ✓ | | | |
| 201 | R | M | 6.3 | 8 | hM4Di | | ✓‡ | | ✓ |
| 207 | R | M | 6.8 | 6 | hM4Di | | ✓‡ | | ✓ |
| 212 | R | F | 3.4 | 8 | hM4Di | ✓ | | | |
| 218 | J | M | 7.2 | 4 | hM4Di | | ✓§ | | |
| 221 | J | M | 6.4 | 4 | hM4Di | ✓ | ✓* | | |
| 225 | J | F | 6 | 7 | hM4Di | ✓ | ✓¶ | | |
| 229 | R | M | 4.6 | 4 | hM4Di | ✓ | ✓** | | |
| 234 | J | M | 5.8 | 5 | hM4Di | ✓ | ✓¶ | | |
| 237 | J | M | 6.9 | 5 | hM4Di | ✓ | | | |
| 238 | J | M | 6.2 | 4 | hM4Di | ✓ | ✓§ | | |
| 245 | J | F | 5.9 | 6 | hM4Di | ✓ | ✓** | | |
| 215 | R | F | 3.4 | 8 | hM3Dq | ✓ | | ✓ | |
| 223 | C | M | 4.2 | 4 | hM3Dq | ✓ | | ✓ | |
| 224 | J | M | 9.9 | 9 | hM3Dq | ✓ | | | ✓ |
| 236 | J | M | 7.1 | 6 | hM3Dq | ✓ | | ✓ | |
| 241 | J | M | 5.1 | 3 | hM3Dq | ✓ | | ✓ | |
| 255 | C | M | 5.2 | 4 | hM3Dq | ✓ | | ✓ | |

Weight and age are the values recorded at the beginning of the experiments.

R, Rhesus; J, Japanese; C, Cynomolgus; M, male; F, female.

*Multi-reward task, #153 and #221, *Oyama et al., 2022a*.

†Delayed-reward task, #163, *Hori et al., 2021*.

‡Delayed matching-to-sample task, #201 and #207, *Hirabayashi et al., 2024*.

§Reversal learning task, #218 and #238, *Oyama et al., 2024*.

¶Brinkman board/foot sensation, #225 and #234, *Hirabayashi et al., 2021*.

**Delayed response task, #229 and #245, *Nagai et al., 2020*.

**Table 2.** Summary of injection location, type of virus vector, titer, and injected volume used in this study.

| ID | Region | Vector | Titer (×10$^{13}$ gc/ml) | Volume (µl) | Symbols in figures 1 | 2 | 5 | S1 |
|----|--------|--------|------|------|---|---|---|----|
| 153 | R-OFC | AAV2-CMV-hM4Di | 1.0 | 54 | | | | |
| | L-rmCD | AAV2-CMV-hM4Di | 2.0 | 3 | | | | |
| 163 | L-dCDh | AAV2.1-hSyn-hM4Di-IRES2-AcGFP | 4.7 | 6 | | | | |
| | R-dCDh | AAV2.1-hSyn-hM4Di-IRES2-AcGFP | 4.7 | 6 | | | | |
| 193 | R-Amygdala | AAV2-CMV-hM4Di | 2.0 | 6 | a | | a | a |
| 201 | R-OFC (microinjected) | AAV2-CMV-hM4Di | 2.2 | 2 | | | | |
| | L-OFC (microinjected) | AAV2-CMV-hM4Di | 2.2 | 2 | | | | |
| 207 | R-OFC (microinjected) | AAV2-CMV-hM4Di | 1.3 | 3 | | | | |
| | L-OFC (microinjected) | AAV2-CMV-hM4Di | 1.3 | 3 | | | | |
| 212 | R-Put | AAV2-CMV-hM4Di | 2.6 | 6 | b | b | b | b |
| 215 | L-Amygdala | AAV2-CMV-hM3Dq | 1.2 | 6 | c | c | c | c |
| 218 | R-OFC | AAV2-CMV-hM4Di | 2.3 | 50 | | | | |
| | L-OFC | AAV2-CMV-hM4Di | 2.3 | 54 | | | | |
| 221 | L-OFC | AAV2-CMV-hM4Di | 2.0 | 50 | | | | |
| | R-rmCD | AAV2-CMV-hM4Di | 2.0 | 3 | | k | k | k |
| 223 | R-Cd | AAV2.1-hSyn-hM3Dq-IRES2-AcGFP | 1.0 | 3 | d | d | d | d |
| | R-Put | AAV2.1-hSyn-hM3Dq-IRES2-AcGFP | 5.0 | 3 | | l | | |
| | L-Cd | AAV2-CMV-hM3Dq-IRES-AcGFP | 1.0 | 3 | | | | |
| | L-Put | AAV2-hSyn-hM3Dq-IRES2-AcGFP | 1.0 | 3 | | | r | r |
| 224 | R-Amygdala | AAV2.1-hSyn-hM3Dq-IRES-AcGFP | 2.0 | 5 | | m | m | m |
| | L-Amygdala | AAV2.1-hSyn-hM3Dq-IRES-AcGFP | 2.0 | 4 | | | s | s |
| 225 | L-SI$_{D2}$ | AAV2-CMV-hM4Di | 1.5 | 4 | | | t | t |
| 229 | R-dlPFC | AAV2.1-hSyn-hM4D-IRES2-AcGFP | 4.7 | 35 | | n | | |
| | L-dlPFC | AAV2.1-hSyn-hM4D-IRES2-AcGFP | 4.7 | 37 | | | | |
| 234 | R-SI$_{D2}$ | AAV2.1-hSyn-hM4D-IRES2-AcGFP | 3.8 | 4 | | o | o | o |
| 236 | R-Amygdala | AAV2-CMV-hM3Dq | 1.2 | 6 | | | | |
| 237 | L-Amygdala | AAV2.1-hSyn-hM4Di-IRES2-AcGFP | 2.0 | 4 | e | e | e | e |
| 238 | R-OFC | AAV2.1-CaMKII-hM4Di-IRES-AcGFP | 1.0 | 49 | | q | | |
| | L-OFC | AAV2.1-CaMKII-hM4Di-IRES-AcGFP | 1.0 | 53 | | p | | |
| 241 | R-Cd | AAV2-hSyn-hM3Dq-IRES-AcGFP | 2.0 | 3 | | | u | u |
| | L-Cd | AAV2-hSyn-hM3Dq | 2.0 | 3 | g | g | g | g |
| | R-Put | AAV2.1-hSyn-hM3Dq-IRES2-AcGFP | 2.0 | 3 | h | h | h | h |
| | L-Put | AAV1-hSyn-hM3Dq-IRES2-AcGFP | 2.0 | 3 | f | f | f | f |
| 245 | R-dlPFC | AAV2.1-hSyn-hM4D-IRES2-AcGFP | 4.7 | 44 | | | | |
| | L-dlPFC | AAV2.1-hSyn-hM4D-IRES2-AcGFP | 4.7 | 40 | | | | |

*Table 2 continued on next page*

*Table 2 continued*

| ID | Region | Vector | Titer (×10¹³ gc/ml) | Volume (μl) | Symbols in figures | | | |
|---|---|---|---|---|---|---|---|---|
| | | | | | 1 | 2 | 5 | S1 |
| 255 | L-VPL | AAV5-hSyn-HA-hM3Dq | 2.7 | 3 | j | j | j | j |
| | R-VPL | AAV2-hSyn-hM3Dq | 1.8 | 3 | i | i | i | i |
| | L-Cd | AAV2-hSyn-hM4Di | 1.2 | 3 | | | x | x |
| | R-Cd | AAV2-hSyn-hM4Di | 1.7 | 3 | | | v | v |
| | L-Put | AAV5-hSyn-hM4Di | 4.6 | 3 | | | | |
| | R-Put | AAV2-hSyn-hM4Di | 1.7 | 3 | | | w | w |

R, right; L, left; OFC, orbitofrontal cortex; rmCD, rostromedial caudate nucleus; dCDh, dorsal part of caudate nucleus head; CD, caudate nucleus; Put, putamen; S1$_{D2}$, hand index finger region of primary somatosensory cortex; dlPFC, dorsolateral prefrontal cortex; VPL, ventral posterolateral nucleus of thalamus.

## Longitudinal assessment of DREADD expression levels

We then conducted a longitudinal assessment of post-peak expression levels for hM4Di and hM3Dq using repeated [¹¹C]DCZ PET measurements from 16 injection sites, including cortical and subcortical regions, across 11 monkeys. *Figure 2* illustrates the temporal changes in normalized ΔBP$_{ND}$, expressed as a percentage of the peak value observed 40–80 days post-AAV injection. hM4Di expression levels remained stable at peak levels for approximately 1.5 years, followed by a gradual decline observed in one case after 2.5 years, and after approximately 3 years in the other two cases (*Figure 2B*, o and b/e, respectively). Compared with hM4Di expression, hM3Dq expression exhibited greater post-peak fluctuations. Nevertheless, it remained at ~70% of peak levels after about 1 year. This post-peak fluctuation was not significantly associated with the cumulative number of DREADD agonist injections (repeated-measures two-way ANOVA, main effect of activation times, $F_{(1,6)} = 5.745$, $P=0.054$). Beyond 2 years post-injection, expression declined to ~50% in one case, whereas another case showed an apparent increase (*Figure 2C*, c and m, respectively).

## DREADDs effectively modulated neuronal activity and behavior for approximately 2 years

We assessed data from 17 monkeys to evaluate the duration over which DREADD activation induced by DREADD agonists produced significant effects on neural activity and behavior. Monkeys expressing hM4Di (N=11) were assessed through behavioral testing, two of which also underwent electrophysiological assessment. Monkeys expressing hM3Dq (N=6) were assessed for changes in glucose metabolism via [¹⁸F]FDG-PET (N=5) or alterations in neuronal activity using electrophysiology (N=1). Across these assessments, significant chemogenetic effects were observed for up to 3 years following AAV vector injection (*Figure 3*). For example, one monkey with bilateral hM4Di expression in the dorsolateral prefrontal cortex (dlPFC; monkey #229) consistently displayed impaired performance on the delayed response task with DCZ administration for up to 2.3 years (867 days) following vector injection (*Oyama et al., 2021*). Similarly, another monkey with hM3Dq expression in the amygdala (monkey #215) exhibited increased glucose metabolism following DCZ administration for up to 2.5 years (926 days) following vector injection, as measured by FDG-PET (*Nagai et al., 2020*).

Although experiment durations varied across monkeys, we consistently observed significant effects throughout the study periods, with only one monkey (#234) being removed from experiments due to a loss of DREADD effectiveness 3 years after transduction. In all other cases, receptor activation was performed repeatedly several tens of times, resulting in no clear loss of efficacy. After the termination of the chemogenetic studies, DREADD expression was verified by in vitro via postmortem immunohistochemical analysis (*Figure 3C–H*). These findings highlight the robust and sustained functionality of DREADD systems in long-term experiments.

As mentioned above, after more than 3 years, we observed the disappearance of hM4Di expression in one monkey (#234), and this was associated with the loss of chemogenetic behavioral effects. Specifically, activating hM4Di in the functionally defined representation of index-finger primary

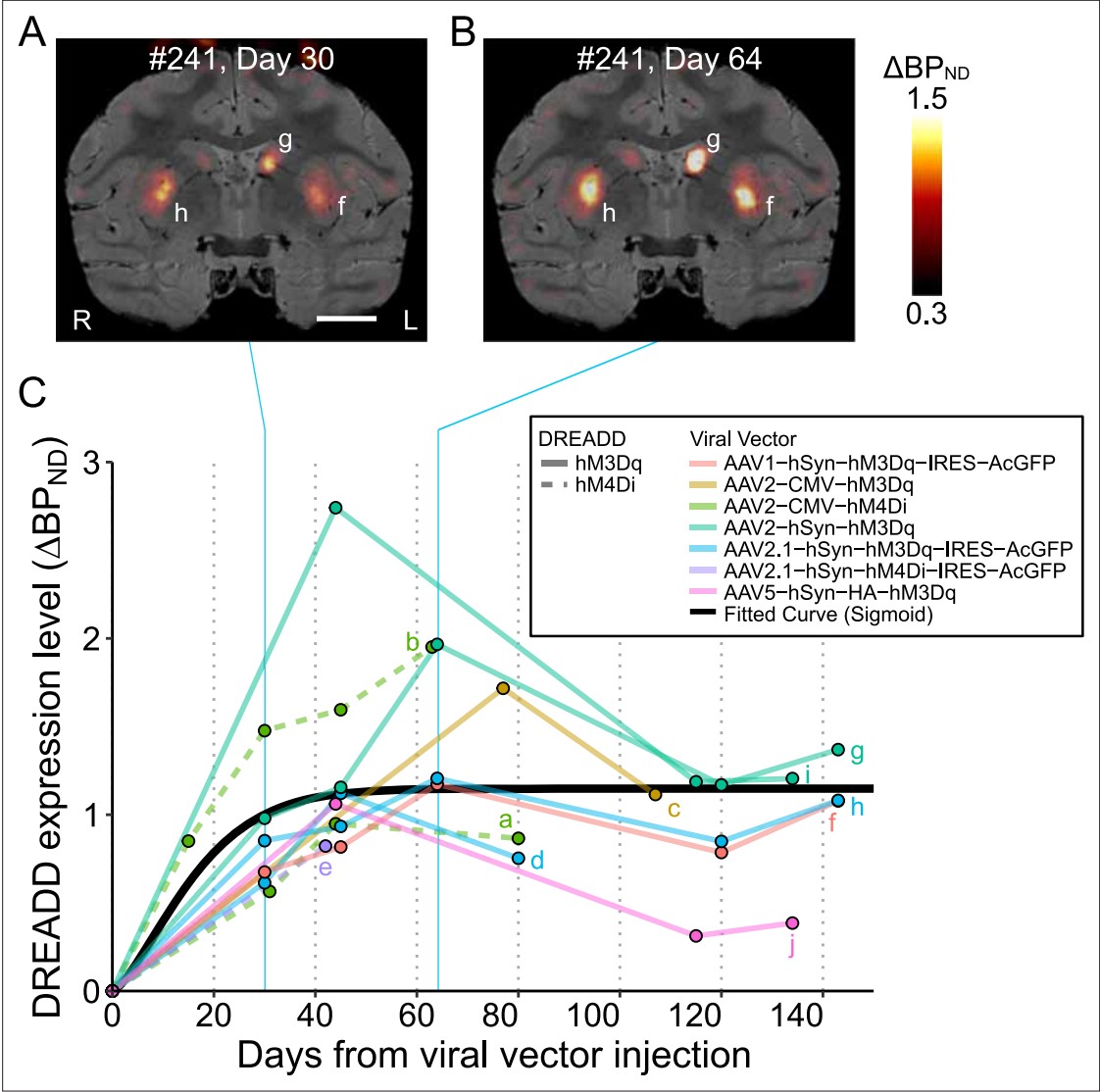

**Figure 1.** Time course of DREADD expression levels up to 150 days after injection. (**A, B**) Parametric coronal images of the increase (relative to baseline) in specific binding of [$^{11}$C]DCZ ($\Delta BP_{ND}$) at 30 (**A**) and 64 (**B**) days after adeno-associated virus (AAV) injection, overlaid on MR images. Images were obtained from a monkey (#241) that received multiple injections of AAV with different constructs (see *Table 2*). (**C**) Time course of in vivo DREADD expression levels ($\Delta BP_{ND}$) up to 150 days after the injections, summarized from 10 regions of interest (ROIs) obtained from seven monkeys. The value at day 0 indicates the baseline (before injection). The black curve is the best-fitted sigmoid curve, which provides a better fit (Bayesian information criterion, BIC = 61.1) than does the double logistic model (BIC = 62.9). Lowercase letters correspond to the DREADD-induced regions described in *Table 2*.

somatosensory cortex (SI$_{D2}$) resulted in reversible behavioral deficits for 2 years, including impaired finger dexterity and hypersensitivity in the foot, confined to the contralateral side (*Figure 4A*; *Hirabayashi et al., 2021*). However, in vivo imaging more than 3 years after injection indicated diminished hM4Di expression, and subsequent behavioral assessments failed to detect any motor deficits or hypersensitivity (*Figure 4B*). Postmortem immunohistochemical examination confirmed the absence of hM4Di, with no evidence of neuronal loss at the injection site (*Figure 4C*).

## Protein tags reduce peak DREADD expression levels

Finally, we investigated the factors influencing the peak level of DREADD expression. To minimize confounding effects related to injection volume and method, this analysis was limited to cases in which AAV vectors were delivered via microinjector (see 'Materials and methods).

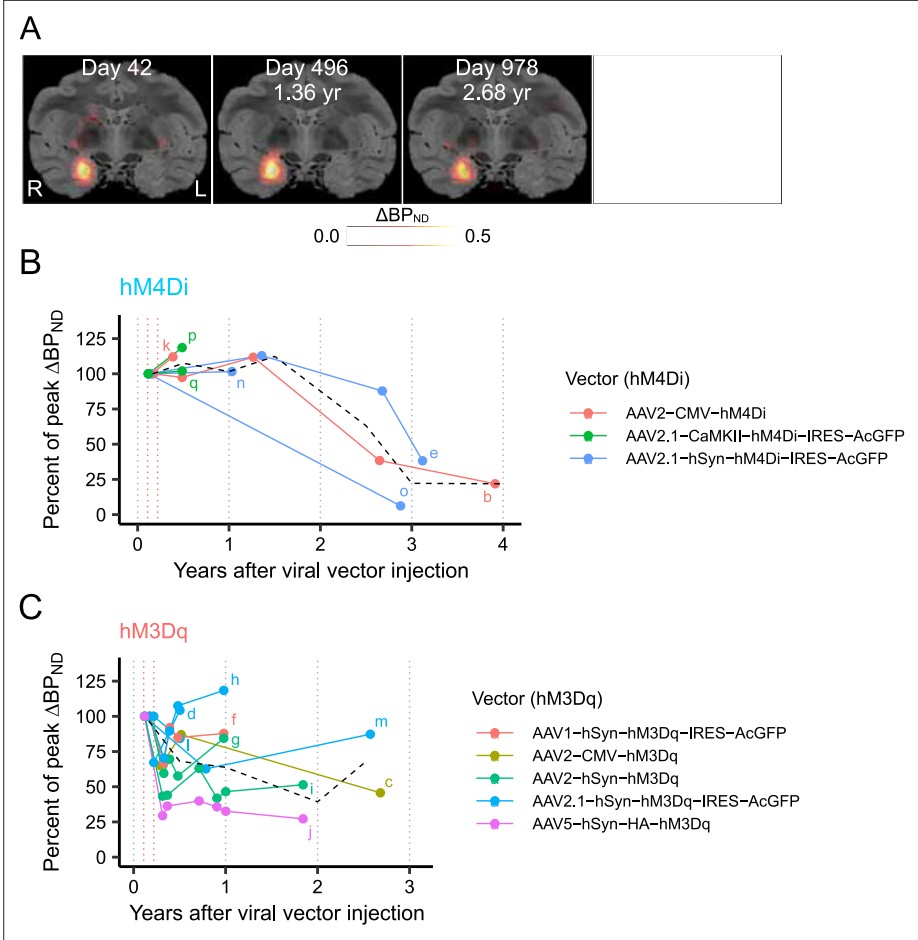

**Figure 2.** Longitudinal time-course of DREADD expression levels. (**A**) Parametric coronal images of DREADD expression levels ($\Delta BP_{ND}$) overlaid on MR images obtained from monkey #237, measured at 42, 496, 978, and 1,139 days after injection. Scale bar is 10 mm. (**B, C**) Time course of hM4Di (**B**) and hM3Dq (**C**) expression levels normalized to the peak $\Delta BP_{ND}$ observed between 40 and 80 days post-injection (time between the dotted red vertical lines). Each colored line represents one injection site, and the lowercase letters at the ends of each line correspond to the injection sites listed in *Table 2*. Each dot represents one [$^{11}$C]DCZ PET measurement. The dashed black line indicates the group average at 6-month intervals. hM4Di expression levels remained stable for approximately 1.5 and those for hM3Dq about 1 year, after which variable patterns of decline were observed.

We applied a linear model incorporating multiple variables, including injection volume, serotype, promoter, titer, tag, and DREADD type, to assess how each factor contributed to peak expression levels. Our analysis revealed that the presence and type of co-expressed protein tags significantly affect peak expression levels ($P<0.008$; *Table 3*). Specifically, peak expression levels were lower for vectors in which the fusion HA-tag sequence was encoded at the 5'-terminal site (5'-HA) of the DREADD sequence than they were for vectors that encoded GFP following the internal ribosome entry site (IRES) sequence, both of which resulted in lower peak expression levels compared with vectors that did not include protein tags (*Figure 5*). A potential interaction between DREADD type and promoter was also observed (*Table 3*, *Figure 5—figure supplement 1*); however, given the limited sample size (n=1 for hM3Dq with CMV), no definitive conclusion can be made.

## Discussion

DREADDs have emerged as invaluable tools in nonhuman primate research due to their ability to exert effects on nonspatially restricted brain regions, allowing for simultaneous and discrete targeting of multiple areas. This capability makes DREADDs promising for investigating the causal roles of specific neural pathways and cell types in the highly specialized brain circuits found in primates.

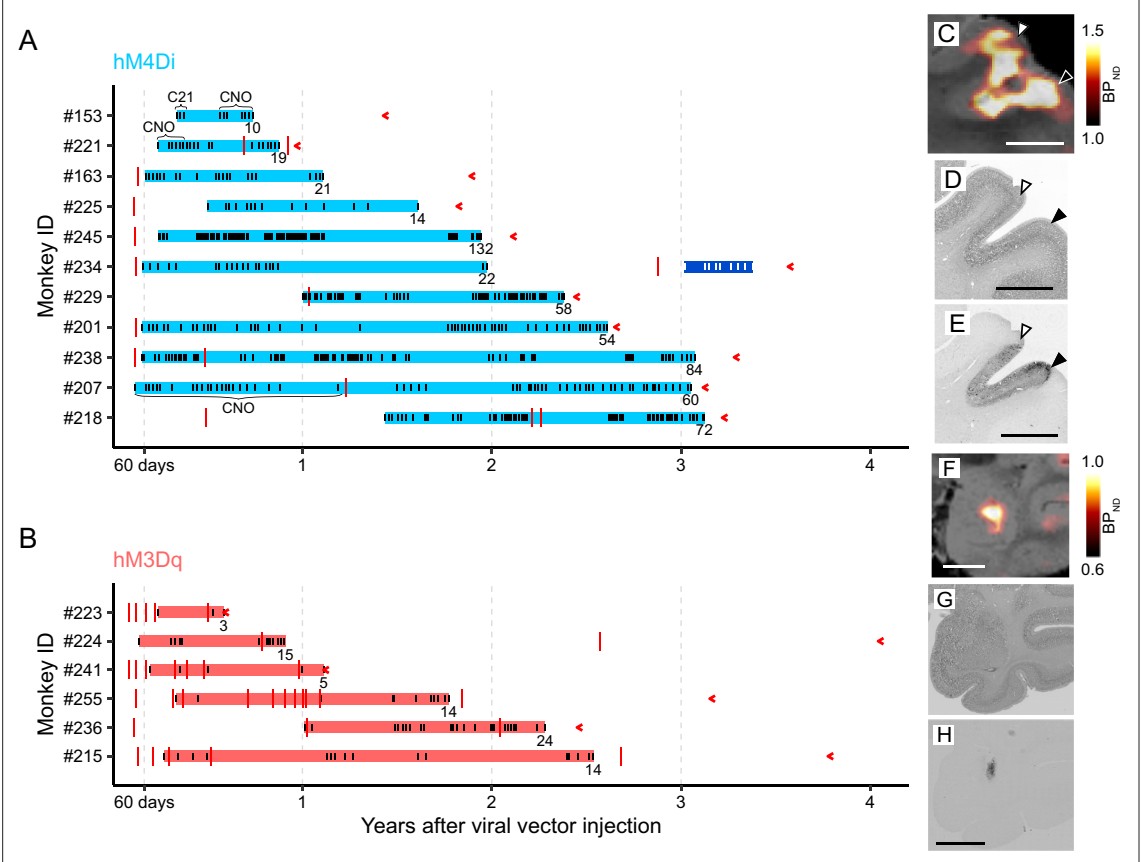

**Figure 3.** Duration of effective functional modulation induced by DREADDs. (**A, B**) Duration of effective functional modulation induced by hM4Di (**A**) and hM3Dq (**B**). Blue and red horizontal bars indicate the duration of successful chemogenetic manipulation. Black and white ticks on the horizontal bars mark the timing of agonist administration, with CNO and Compound 21 (C21) explicitly stated; all other ticks correspond to DCZ. For example, repeated DCZ administration consistently induced behavioral change in monkey #229 and increased glucose metabolism in monkey #215. The corresponding functional assessments are summarized in *Table 1*. The dark blue bar indicates the duration of failed chemogenetic manipulation in monkey #234 (see also *Figure 4*). Total numbers of agonist administrations (DREADD activation) are shown at the end of each duration bar. Red ticks indicate the timing of DCZ-PET scans and red arrowheads indicate the timing of perfusion. (**C**) In vivo visualization of hM4Di expression in the dorsolateral prefrontal cortex 377 days after injection (monkey #229). The coronal PET image showing specific binding of [¹¹C]DCZ is overlaid on MR images. (**D, E**) Nissl- (**D**) and DAB-stained (**E**) sections corresponding to (**C**), representing immunoreactivity against reporter protein. White and black arrowheads represent the dorsal and ventral borders of the target regions, respectively. Scale bars, 5 mm. (**F**) In vivo visualization of hM3Dq expression in the amygdala 980 days after the injection (monkey #215). The coronal PET image showing specific binding of [¹¹C]DCZ is overlaid on MR images. (**G, H**) Nissl- (**G**) and DAB-stained (**H**) sections corresponding to (**F**), representing immunoreactivity against reporter protein.

However, ethical and logistical constraints on using a large number of monkeys highlight the need of moving beyond a trial-and-error approach. It is essential to aggregate data and provide insights that are more widely applicable, thus enabling the development of a more efficient experimental design for DREADD research in monkeys. In vivo PET visualization of DREADDs offers a noninvasive and quantitative means for real-time monitoring of receptor expression. The present study analyzed PET and functional/behavioral data from 43 injection sites across 20 monkeys and revealed three key findings: (1) after AAV injection, DREADD expression levels peaked after approximately 60 days and remained stable for up to 1.5 years, followed by a gradual decline observed after 2–3 years; (2) significant chemogenetic effects on neural activity and behavior persisted for around 2 years; and (3) protein tags significantly influenced peak expression levels, with co-expressed protein tags reducing peak expression. These results offer quantitative insights into the temporal dynamics of DREADD expression, thereby informing strategies for designing long-term experiments. They also confirm the efficacy and reliability of DREADD-mediated interventions.

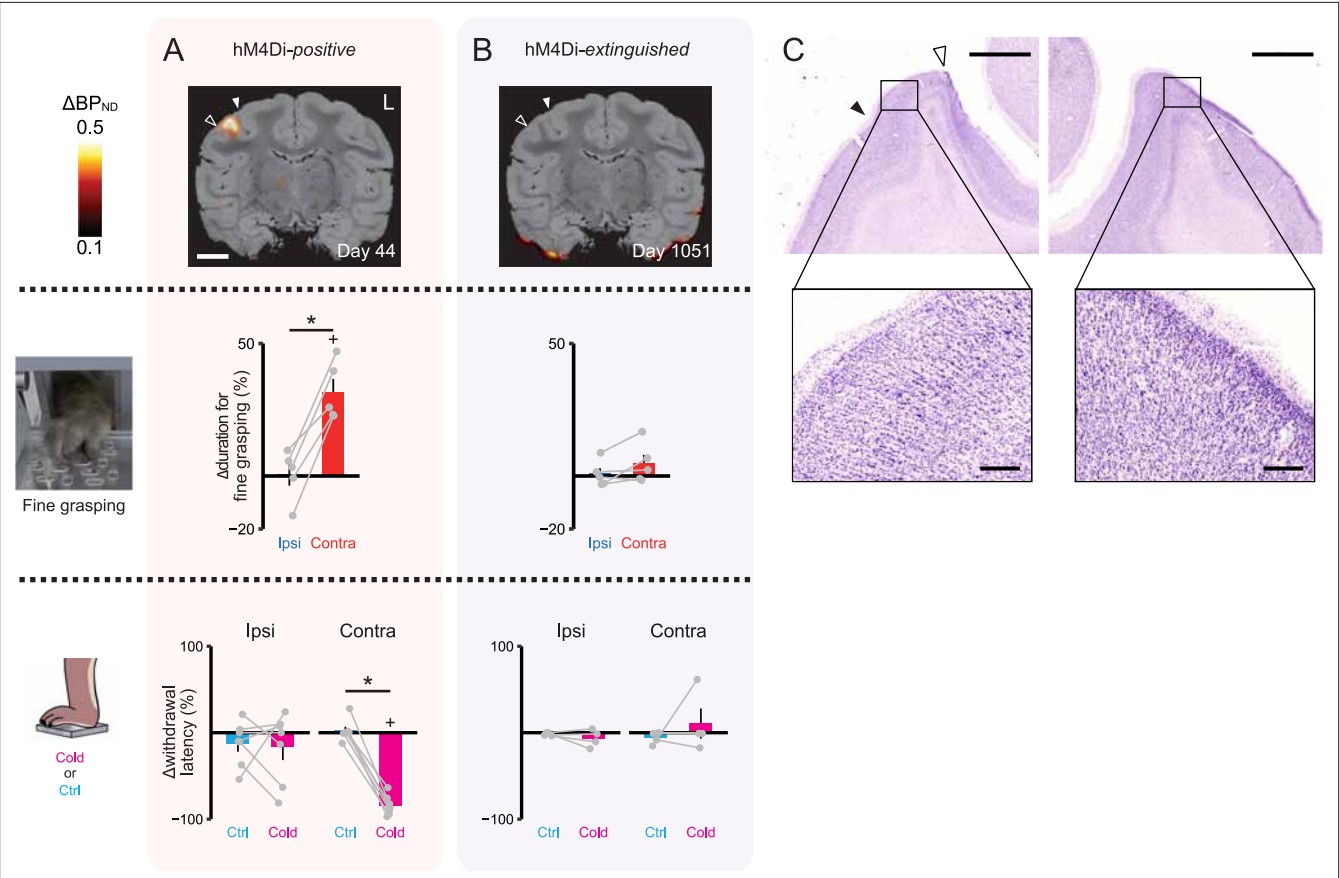

**Figure 4.** Disappearance of behavioral effects in an hM4Di-extinguished monkey (#234). (**A**) Top: coronal PET image showing ΔBP$_{ND}$ of [$^{11}$C]DCZ data overlaid on an MR image, obtained 44 days after viral vector injection into the right SI$_{D2}$. Filled and open arrowheads represent the central and ipsilateral sulci, respectively. Middle: performance of fine grasping using a modified Brinkman board task, assessing the monkey's ability to pick up small food pellets with its thumb and index fingers. Values indicate the change in total duration to complete the task between pre- and post-DCZ administration sessions. *Ipsi* (*Contra*) refers to performance using the hand ipsilateral (contralateral) to the hM4Di-expressing SI$_{D2}$. Gray lines connect performances from the same sessions using different hands. Error bars, s.e.m. *$P<0.002$, paired *t*-test. +$P<0.004$, paired *t*-test, adjusted for multiple comparisons. n=5 sessions. Bottom: DCZ-induced change in the foot withdrawal latency in response to cold (magenta) or control (cyan) stimulation. Negative values indicate faster withdrawal latency following DCZ administration. Gray lines connect performances from the same sessions. Error bars, s.e.m. *$P<0.001$, paired *t*-test. +$P<0.001$, paired *t*-test comparing between pre- and post-DCZ administration. n=7 sessions. (**B**) Same types of measurements as in (**A**), but following the extinction of hM4Di expression. The PET image was obtained about 3 years (1051 days) post-vector injection. n=5 and n=4 sessions for Brinkman board task and foot withdrawal test, respectively. (**C**) A Nissl-stained section demonstrating the absence of neuronal loss at the vector injection sites (left panel) and the contralateral side (right panel). The locations of the filled and open arrowheads correspond to those shown in (**A**) and (**B**). Scale bars are 2.5 mm and 250 μm.

## Technical considerations

In vivo monitoring of gene expression has been conducted using fluorescent reporters observed through a window or fiber, providing a method to verify transducing protein expression and localization (*Diester et al., 2011*; *Ruiz et al., 2013*). Although useful for optogenetics and imaging studies, these methods are invasive, less quantitative, and spatially limited, often requiring cranial surgeries and capturing only two-dimensional data. In contrast, DCZ PET imaging provides a direct, noninvasive, volumetric measure of receptor expression, making it particularly suitable for studies in large animals such as macaques and for translational research with potential application to future human therapies. Nevertheless, PET measurement has certain limitations, including potential variability due to individual differences, underestimation of DREADD expression levels due to the partial volume effect, and the influence of anesthesia on kinetic parameters during imaging sessions. In this study, we have considered these factors and minimized their impact by applying appropriate ROI placements and excluding data collected under inadequate anesthetic conditions (see 'Materials and methods').

**Table 3.** Results of the linear model analysis.

| Effect | DFn | DFd | F-value | P value | Effect size ($\eta^2$G) |
|---|---|---|---|---|---|
| Titer | 1 | 4 | 2.093 | 0.222 | 0.343 |
| DREADD | 1 | 4 | 2.565 | 0.184 | 0.391 |
| Promoter | 1 | 4 | 5.280 | 0.083 | 0.569 |
| Tag | 2 | 4 | 19.84 | 0.008* | 0.908 |
| Serotype | 3 | 4 | 2.105 | 0.242 | 0.612 |
| Volume | 1 | 4 | 0.090 | 0.779 | 0.022 |
| Titer:promoter | 1 | 4 | 2.346 | 0.200 | 0.370 |
| DREADD:promoter | 1 | 4 | 10.74 | 0.031* | 0.729 |
| Titer:tag | 2 | 4 | 1.494 | 0.328 | 0.428 |
| Titer:volume | 1 | 4 | 3.199 | 0.148 | 0.444 |
| DREADD:volume | 1 | 4 | 1.497 | 0.288 | 0.272 |

The ANOVA table describes the factors contributing to the level of expression. The optimal model, selected based on Akaike's information criterion, was the following: $\Delta BP_{ND}$ = viral titer + DREADD type + promoter + reporter tag + serotype + injection volume + viral titer:promoter + DREADD type:promoter + viral titer:reporter-tag + viral titer:injection volume + DREADD type:injection volume.

DFn, degrees of freedom numerator; DFd, degrees of freedom denominator; $\eta^2$G, generalized eta-squared.

*$P<0.05$.

This study included a retrospective analysis of datasets pooled from multiple studies conducted within a single laboratory, which inherently introduced variability across injection parameters and scan intervals. While such an approach reflects real-world practices in long-term NHP research, future studies, including multicenter efforts using harmonized protocols, will be valuable for systematically assessing inter-individual differences and optimizing key experimental parameters.

## Temporal dynamics of DREADD expression

Our [$^{11}$C]DCZ PET measurement showed that DREADD expression peaked around 60 days after AAV injection, regardless of whether hM4Di or hM3Dq was used. This pattern was consistent with our previous data regarding hM4Di expression visualized using [$^{11}$C]clozapine (*Nagai et al., 2016*). This gradual rise in AAV-mediated gene expression likely reflects rate-limiting steps following viral infection, including transport to the nucleus, uncoating, and conversion to double-strand DNA (*Duan et al., 2000*; *Ferrari et al., 1996*; *Li and Samulski, 2020*; *Wang et al., 2007*). Similar temporal patterns of AAV-mediated gene expression have been observed in previous studies using in vivo fluorescent measurement. For example, in rats transfected with an AAV5 encoding fluorescence protein (EYFP) in motor cortex and the hippocampus, in vivo fluorescence signals increased sigmoidally, rapidly until day 35, then slowed down (*Diester et al., 2011*). Similarly, in macaques, ChR2 expression monitored via fluorescence intensity peaked at around 60 days after AAV9 injection (*Nakamichi et al., 2019*).

Although we modeled the time course of DREADD expression using a single sigmoid function, PET data from several monkeys showed a modest decline following the peak. While the sigmoid model captured the early-phase dynamics and offered a reliable estimate of peak timing, additional PET scans—particularly between 60 and 120 days post-injection—will be essential to fully characterize the biological basis of the post-peak expression trajectories.

Despite being derived from a limited dataset, our findings offer important insights for designing chemogenetics studies in primates. Specifically, beginning functional experiments at approximately 60 days post-AAV injection will yield reliable results and avoid unnecessary delays. This timeline may also be applicable to other genetic studies in monkeys, such as those using optogenetics or calcium imaging, which also require proper expression of functional proteins. However, while these tools rely on rapid, light-driven responses and are relatively less dependent on stable functional protein expression, DREADDs require sustained expression for consistent long-term effects.

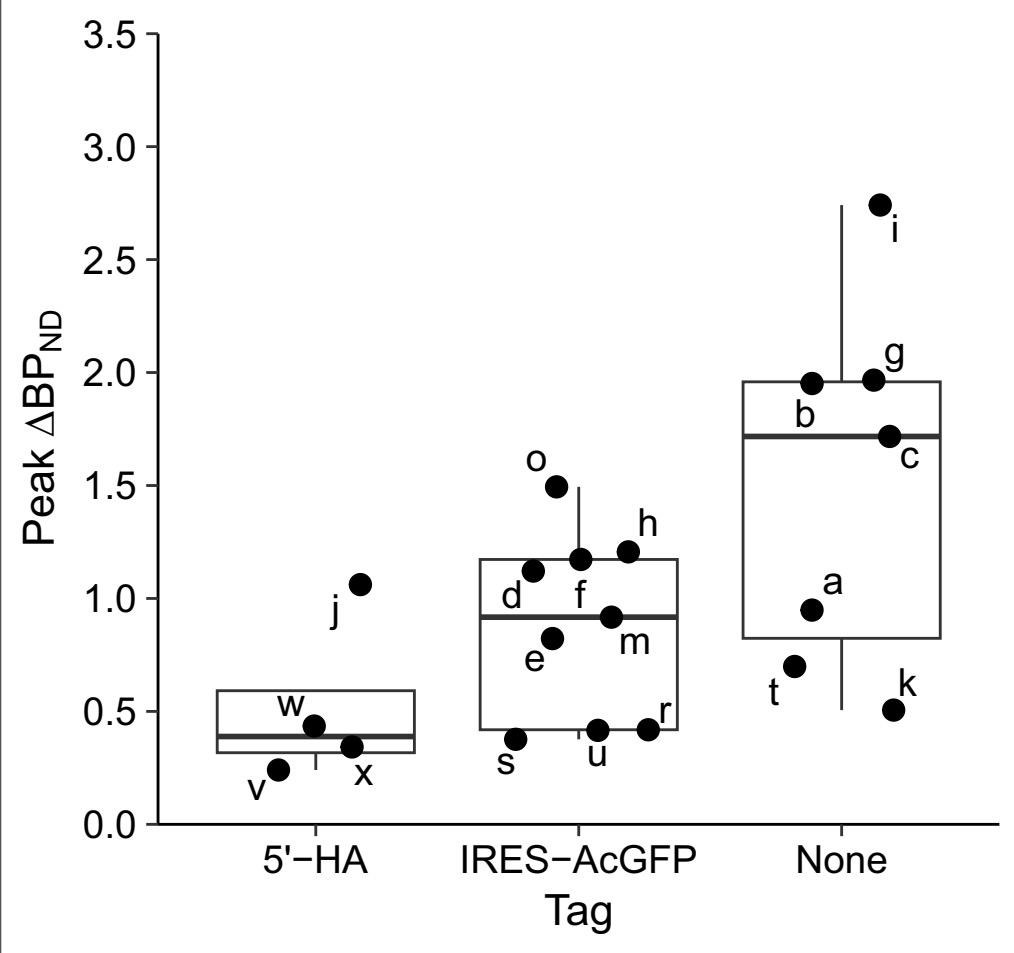

**Figure 5.** Effect of protein tags on peak DREADD expression levels. Peak expression levels ($\Delta BP_{ND}$) for different constructs showing the presence and type of protein tag. Box plots represent the median (central line), interquartile range (box), and the range (whiskers) of the data. Individual data points are labeled with lowercase letters and correspond to the injection sites listed in *Table 2*. Note that two cases (right putamen of #223 and left putamen of #255) were identified as outliner and excluded from the analysis (see 'Materials and methods').

The online version of this article includes the following figure supplement(s) for figure 5:

**Figure supplement 1.** Effect of promoter and DREADD type on peak expression levels.

## Factors influencing peak DREADD expression

Although higher AAV titers generally result in increased transgene expression in neurons, excessively high titers can trigger immune responses. In the current study, we used titers that ranged from 1 to $5 \times 10^{13}$ gc/ml, achieving a balance between effective expression and minimal immune activation. This may explain why virus titer did not emerge as a significant factor affecting expression levels in our analysis. By contrast, our analysis indicated that the presence and type of protein tags significantly influenced peak DREADD expression levels. Vectors encoding DREADDs with protein tags, particularly 5'-terminal HA tags, were associated with a reduction of peak expression levels. Thus, although these tags are commonly used for antibody detection in DREADD constructs, they negatively impact expression efficacy and potentially affect protein function. Our findings are consistent with a previous PET study using [$^{11}$C]clozapine and immunohistochemical analysis, which also found reduced hM4Di expression in constructs with 5'-terminal HA tags (*Nagai et al., 2016*). Additionally, constructs incorporating IRES-GFP might also result in lower expression levels due to the size and composition of the AAV vector, as has been shown in previous studies (*Furler et al., 2001*; *Kimura et al., 2023*; *Mizuguchi et al., 2000*; *Zhou et al., 1998*). Nevertheless, these constructs have been frequently used in rodent research and have also been effective in producing long-term chemogenetic effects in monkey

studies, including the current one (e.g., monkeys #229 and #238). Thus, while the observed decline in DREADD expression resulting from these constructs is not inherently detrimental to experimental outcomes in monkey studies, it serves as a warning that we must be careful when designing vectors to avoid reductions in experimental efficacy.

## Implications for long-term experiments with DREADD expression

One of the most important findings was that AAV-mediated DREADD expression was maintained for up to 2 years for hM4Di and at least 1 year for hM3Dq. Throughout the duration of the experiments, chemogenetic effects remained consistent, as verified by individual behavioral and functional assessments. However, hM4Di expression diminished after 3 years, coinciding with the loss of chemogenetic behavioral effects in at least one case. This indicates that while chemogenetic effects can be maintained for up to 2 years, they likely decline gradually beyond this period. Consequently, researchers should anticipate consistent chemogenetic effects for a maximum of 2 years following AAV injection and should be mindful that longer-term experiments are at risk for a decline in effectiveness. Specifically, the inevitable failure to detect DREADDs via immunohistochemistry highlights an important caveat for researchers: timely termination of experiments to ensure verification through in vitro examination is necessary.

The gradual decline in expression levels observed beyond 2 years has several potential causes. First, AAV vectors typically remain episomal in transduced cells, without integrating into the host genome (*Weitzman and Linden, 2011*). While this poses less of an issue in nondividing neurons, episomal DNA can still be subject to gradual loss or degradation over time. Second, several mechanisms may contribute to the gradual downregulation of transgene expression, including epigenetic modifications, immune responses, and genome stability (*Muhuri et al., 2022*). For example, promoter silencing—often associated with epigenetic modifications—has been reported (*Gray et al., 2011*). Neuron-specific promoters have been shown to achieve more efficient and prolonged transgene expression than ubiquitous promoters like CMV (*Paterna et al., 2000*). Our study found no evidence of cell death at AAV injection sites (*Figure 4C*), suggesting that neuronal loss was not a primary cause of the eventual decline in expression levels. Repeated receptor activation did not affect DREADD expression levels or chemogenetic efficacy in a previous study (*Oyama et al., 2022b*). Thus, identifying the factors that cause expression to eventually decline remains of critical importance. With this knowledge, we can develop strategies to maintain persistent transgene expression in neurons for more than 2 years, which will benefit both chemogenetic studies and therapeutic applications. Despite this uncertainty, our data offers important guidelines for chemogenetics studies in nonhuman primates, demonstrating that up to 2 years of reliable expression is achievable.

## Translational implications

This research has significant translational implications, particularly for the development of DREADD-based therapies for treating neurological and psychiatric disorders. DREADD-based approaches have shown promising therapeutic potential, with successful application in monkey models of disease conditions such as epileptiform seizures (*Miyakawa et al., 2023*) and Parkinson's disease (*Chen et al., 2023*). Extending these findings to human clinical applications positions PET imaging as a powerful tool for real-time, noninvasive confirmation of target protein expression—a critical factor for ensuring therapeutic efficacy. PET reporter imaging has already shown its value in human gene therapy for certain movement disorders, where sustained protein expression has been correlated with treatment efficacy (*Mittermeyer et al., 2012*; *Tai et al., 2022*). Our study reinforces this utility by demonstrating that when DREADD expression was no longer detectable via PET, the chemogenetic effects were also diminished.

## Conclusion

Our study describes short- and long-term dynamics of DREADD expression in nonhuman primates using [$^{11}$C]DCZ PET imaging, complemented by assessment of functional efficacy. The findings provide critical insights into the practicality of multi-year studies using chemogenetic neuromodulation with DREADDs, including the expression time course and factors affecting expression stability. This work offers valuable guidance for designing future long-term nonhuman primate studies and underscores

the translational potential of using DREADDs combined with PET imaging for noninvasive, gene-targeted therapies for neurological and psychiatric disorders.

## Materials and methods

### Subjects

A total of 20 macaque monkeys (10 Japanese, *Macaca fuscata*; 8 Rhesus, *Macaca mulatta*; 2 Cynomolgus, *Macaca fascicularis*; 16 males, 4 females; weight, 3.4–9.9 kg; age, 3–13 years at the beginning of experiments) were used (*Table 1*). The monkeys were kept in individual primate cages in an air-conditioned room. A standard diet, supplementary fruits/vegetables, and a tablet of vitamin C (200 mg) were provided daily. All experimental procedures involving animals were carried out in accordance with the Guide for the Care and Use of Nonhuman Primates in Neuroscience Research (The Japan Neuroscience Society; https://www.jnss.org/en/animal_primates) and were approved by the Animal Ethics Committee of the National Institutes for Quantum Science and Technology (Permit Number: 11–1038).

### Viral vector production

We used a total of 11 AAV vectors. AAV2-Syn-HA-hM3Dq, AAV2-Syn-HA-hM4Di, and AAV5-Syn-HA-hM4Di were purchased from Addgene (MA, USA). The other viral vectors were produced by a helper-free triple transfection procedure, which was purified by affinity chromatography (GE Healthcare, Chicago, USA). Viral titer was determined by quantitative polymerase chain reaction using Taq-Man technology (Life Technologies, Waltham, USA).

### Surgical procedures and viral vector injections

Surgical procedures have been described in detail in previous reports (*Hirabayashi et al., 2021*; *Nagai et al., 2020*; *Nagai et al., 2016*; *Oyama et al., 2023*; *Oyama et al., 2022b Oyama et al., 2021*). The AAV vectors and brain regions into which they were injected are summarized in *Table 2*. Before surgery, structural scans of the head were acquired via magnetic resonance (MR) imaging (7 tesla 400 mm/SS system, Bruker, Billerica, MA, USA) and X-ray computed tomography (CT) (Accuitomo170, J. MORITA CO., Kyoto, Japan) under anesthesia (continuous intravenous infusion of propofol 0.2–0.6 mg/kg/min). Overlay MR and CT images were created using PMOD image analysis software (PMOD 4.4, PMOD Technologies Ltd, Zurich, Switzerland) to estimate the stereotaxic coordinates of target brain structures.

Surgeries were performed under aseptic conditions in a fully equipped operating suite. We monitored body temperature, heart rate, peripheral oxygen saturation ($SpO_2$), and end-tidal $CO_2$ throughout all surgical procedures. Monkeys were immobilized by intramuscular (i.m.) injection of ketamine (5–10 mg/kg) and xylazine (0.2–0.5 mg/kg) after i.m. injection of atropine sulfate (0.02–0.05 mg/kg) and then intubated with an endotracheal tube. Anesthesia was maintained with isoflurane (1%–3%, to effect). After surgery, prophylactic antibiotics and analgesics (cefmetazole, 25–50 mg/kg/day; ketoprofen, 1–2 mg/kg/day) were administered for 7 days.

Viral vectors were injected with a stereotaxic apparatus or a hand-held syringe. For stereotaxic injection, burr holes (~8 mm in diameter) were created for inserting the injection needle. Viruses were pressure-injected using a 10 µl Hamilton syringe (model 1701 RN, Hamilton, Reno, UV) mounted on a motorized microinjector (UMP3T-2, WPI, Sarasota, FL) held by a manipulator (model 1460, David Kopf, Tujunga, CA) on the stereotaxic frame. After making an approximately 3 mm cut in the dura mater, the injection needle was inserted into the brain and slowly moved down 1–2 mm beyond the target and then kept stationary for 5 min, after which it was pulled back up to the target location. Injection speed was 0.1–0.5 µl/min. After each injection, the needle remained in situ for 15 min to minimize backflow along the needle and then was pulled out very slowly.

For handheld injection, the target cortex was exposed by removing a bone flap and reflecting the dura mater. Injections were made under visual guidance through an operating microscope (Leica M220, Leica Microsystems GmbH, Wetzlar, Germany) at an oblique angle to the brain surface. Each hemisphere received nine tracks of injections: one at the caudal tip, four along the dorsal bank, and four along the ventral bank of the principal sulcus, with 3–5 µL per track, depending on the depth (*Oyama et al., 2023*).

## PET imaging

PET imaging was conducted as previously reported (*Nagai et al., 2020*). Briefly, PET scans were performed using a microPET Focus 220 scanner (Siemens Medical Solutions USA, Malvern, USA). Monkeys were initially immobilized with ketamine (5–10 mg/kg) and xylazine (0.2–0.5 mg/kg) and maintained under anesthesia with isoflurane (1–3%) throughout the PET procedures. The isoflurane level was carefully adjusted to ensure the stability of continuously monitoring physiological parameters, such as body temperature, heart rate, $SpO_2$, and end-tidal $CO_2$. End-tidal $CO_2$ levels were maintained between 35 and 45 mmHg to prevent substantial changes in cerebral blood flow (*Markwalder et al., 1984*), which could impact tracer kinetics. Transmission scans were performed for about 20 min with a Ge-68 source. Emission scans were acquired in 3D list mode with an energy window of 350–750 keV after intravenous bolus injection of [$^{11}$C]DCZ (~350 MBq) and [$^{18}$F]FDG (~200 MBq). Data acquisition lasted for 90 min.

To estimate the specific binding of [$^{11}$C]DCZ, regional binding potential relative to a nondisplaceable radioligand ($BP_{ND}$) was calculated by PMOD with an original multilinear reference tissue model (MRTMo) (*Yan et al., 2021*). The cerebellum was used as a reference region, and t* was 15 min. The DREADD expression level was defined as $\Delta BP_{ND}$, calculated by subtracting the pre-injection value from the post-injection value because the $BP_{ND}$ value included off-target binding. This minimized individual differences because off-target binding varied among subjects. The volumes of interest (VOIs) reflecting DREADD expression were identified by drawing contours at half of the maximum $\Delta BP_{ND}$ value in the $\Delta BP_{ND}$ parametric image, which was smoothed with a 2 mm Gaussian filter, and overlaid onto the corresponding region in each individual MR image.

For the FDG study, DCZ (1 µg/kg) or vehicle was administered intravenously 1 min before FDG injection. Data were converted to standardized uptake value (SUV) images, averaged between 30- and 60 min, and normalized to SUV ratio (SUVR) images using the mean whole-brain value. Metabolic changes induced by DCZ were calculated as $\Delta$SUVR and defined by the different DCZ and vehicle conditions, with VOIs for FDG analysis matching those used in the [$^{11}$C]DCZ-PET study.

## Administration of DREADD agonists

Deschloroclozapine (DCZ; HY-42110, MedChemExpress) was the primary agonist used. DCZ was first dissolved in dimethyl sulfoxide (DMSO; FUJIFILM Wako Pure Chemical Corp.) and then diluted in saline to a final volume of 1 ml, with the final DMSO concentration adjusted to 2.5% or less. DCZ was administered intramuscularly at a dose of 0.1 mg/kg for hM4Di activation, and at 1–3 µg/kg for hM3Dq activation. For behavioral testing, DCZ was injected approximately 15 min before the start of the experiment unless otherwise noted. Fresh DCZ solutions were prepared daily.

In a limited number of cases, clozapine-N-oxide (CNO; Toronto Research Chemicals) or compound 21 (C21; Tocris) was used as an alternative DREADD agonist for some hM4Di experiments. Both compounds were dissolved in DMSO and then diluted in saline to a final volume of 2–3 ml, also maintaining DMSO concentrations below 2.5%. CNO and C21 were administered intravenously at doses of 3 mg/kg and 0.3 mg/kg, respectively.

## Behavioral tasks

The behavioral experiments are summarized in *Table 1*. Complete descriptions of the detailed procedures for each behavioral experiment are available in previous reports (*Hirabayashi et al., 2024*; *Hirabayashi et al., 2021*; *Hori et al., 2021*; *Oyama et al., 2024*; *Oyama et al., 2022a*; *Oyama et al., 2021*). Here we focus on a modified Brinkman board task and a sensitivity to cold stimulation task, which were used to analyze the behavior of a monkey (#234) that had hM4Di expressed in its primary somatosensory cortex (*Figure 4*). In the modified Brinkman board task, monkeys picked food pellets out of 10 slots in a board with one of their hands (i.e., contralateral or ipsilateral to the DREADD-expressing hemisphere), and the amount of time needed to pick up all 10 pellets was recorded. The time to complete each trial was normalized to the number of pellets successfully picked up to obtain a value that reflected manual dexterity. In a single experimental session, a monkey performed five to ten trials with each hand before and 10 min after intravenous systemic administration of DCZ. In the sensitivity to cold task, monkeys were trained to place a sole of one of their feet on a cold metal plate that was placed in front of their monkey chair. The latency with which they withdrew their foot from the plate was recorded and used as an index of sensitivity to a cutaneous (cold) stimulus. If a monkey did

not withdraw its foot within 60 s, the trial was terminated and a withdraw latency of 60 s was assigned for that trial. The monkeys performed five to ten trials for each of two temperature conditions (30°C for control and 10°C for cold stimulation) both before and 10 min after DCZ administration. Chemogenetic effects were assessed by comparing performance on the tasks pre- and post-injection and between contralateral and ipsilateral hands and feet (only the contralateral side should be affected by DCZ).

## Data analysis and statistics

All data and statistical analyses were performed using the R statistical computing environment (*R Development Core Team, 2025*). To model the time course of DREADD expression, we used a single sigmoid function, referencing past in vivo fluorescent measurements (*Diester et al., 2011*). Curve fitting was performed using least squares minimization. For comparison, a double logistic function was also tested and evaluated using the Bayesian information Criterion (BIC) to assess model fit. Repeated-measures two-way ANOVAs were used to examine the effect of cumulative number of activation times ×days after injection on post-peak fluctuation.

To identify factors influencing peak expression levels, a stepwise model selection using the stepAIC() function from the MASS package (*Venables and Ripley, 2002*) was performed. To minimize potentially confounding effects related to injection method and volume, we restricted this analysis to cases in which a microinjector was used to deliver the AAV and excluded cortical injections into the orbitofrontal cortex and dlPFC, which involved larger, manually administered injection volumes. Starting with a full model that included six factors—virus titer, DREADD type, promoter, tag, serotype, and injection volume—along with all their interactions, the best-fitting linear model was determined based on Akaike information criterion (AIC). A permutation-based outlier analysis was also performed to detect and exclude statistical outliers that could disproportionally influence the regression results. The resulting best model included six variables and five interactions (Titer:Promoter, DREADD:Promoter, Titer:Tag, Titer:Volume, and DREADD:Volume). The significance of their contributions to the peak expression level, as well as the effect sizes (generalized eta-squared, $\eta^2G$), were assessed using type II analysis of variance with the anova_test() function from the rstatix package (*Kassambara, 2023*).

## Histology and immunostaining

For histological inspection, monkeys were deeply anesthetized with sodium pentobarbital (50 mg/kg, i.v.) or sodium thiopental (50 mg/kg, i.v.) after administration of ketamine hydrochloride (5–10 mg/kg, i.m.) and then transcardially perfused with saline at 4°C, followed by 4% paraformaldehyde in 0.1 M phosphate buffered saline (PBS, pH 7.4) at 4°C. The brain was removed from the skull, postfixed in the same fresh fixative overnight, saturated with 30% sucrose in phosphate buffer (PB) at 4°C, and then cut serially into 50-µm-thick sections on a freezing microtome. For visualizing immunoreactive GFP (co-expressed with hM4Di) signals, a series of every sixth section was immersed in 1% skim milk for 1 h at room temperature and incubated overnight at 4°C with rabbit anti-GFP monoclonal antibody (1:500, G10362, Thermo Fisher Scientific, Waltham, MA, USA) in PBS containing 0.1% Triton X-100 and 1% normal goat serum for 2 days at 4°C. The sections were then incubated in the same fresh medium containing biotinylated goat anti-rabbit IgG antibody (1:1000; Jackson ImmunoResearch, West Grove, PA, USA) for 2 h at room temperature, followed by avidin-biotin-peroxidase complex (ABC Elite, Vector Laboratories, Burlingame, CA, USA) for 2 h at room temperature. For visualizing the antigen, the sections were reacted in 0.05 M Tris-HCl buffer (pH 7.6) containing 0.04% diaminobenzidine (DAB), 0.04% $NiCl_2$, and 0.003% $H_2O_2$. The sections were mounted on gelatin-coated glass slides, air-dried, and cover-slipped. Parts of other sections were Nissl-stained with 1% Cresyl violet. Images of sections were digitally captured using an optical microscope equipped with a high-grade charge-coupled device camera (Biorevo, Keyence, Osaka, Japan).

The results indicate that the reporter-tag and the interactions between DREADD type:promoter significantly affected DREADD expression levels. However, interpretation of the latter should be made with caution due to limited sampling in some combinations (see *Figure 5—figure supplement 1* and main text).

## Acknowledgements

This study was supported by MEXT/JSPS KAKENHI grant numbers JP19K08138, JP23K27098 (to YN), JP24H00734 (to TH), JP22H05157 (to KI), JP20H05955, and JP24H00069 (to TM), JST PRESTO grant numbers JPMJPR22S3 (to KO), AMED grant numbers JP23wm0625001 (to TM) and JP24wm0625307 (to TH), and the Moonshot Research & Development Program (Millennia Program) from JST grant number JPMJMS2295 (to TM and KI). Japanese monkeys were provided by the Japan MEXT National Bio-Resource Project 'Japanese Monkeys'. We thank Jun Kamei, Ryuji Yamaguchi, Yuichi Matsuda, Yoshio Sugii, Takashi Okauchi, Risa Suma, Tomomi Kokufuta, Rie Yoshida, and Akari Ueshiba for their technical assistance.

## Additional information

### Funding

| Funder | Grant reference number | Author |
| --- | --- | --- |
| Japan Society for the Promotion of Science | JP19K08138 | Yuji Nagai |
| Japan Society for the Promotion of Science | JP23K27098 | Yuji Nagai |
| Japan Society for the Promotion of Science | JP24H00734 | Toshiyuki Hirabayashi |
| Japan Society for the Promotion of Science | JP22H05157 | Ken-ichi Inoue |
| Japan Society for the Promotion of Science | JP20H05955 | Takafumi Minamimoto |
| Japan Society for the Promotion of Science | JP24H00069 | Takafumi Minamimoto |
| Japan Science and Technology Agency | 10.52926/jpmjpr22s3 | Kei Oyama |
| Japan Agency for Medical Research and Development | JP23wm0625001 | Takafumi Minamimoto |
| Japan Agency for Medical Research and Development | JP24wm0625307 | Toshiyuki Hirabayashi |
| Japan Science and Technology Agency | 10.52926/jpmjms2295 | Ken-ichi Inoue Takafumi Minamimoto |

The funders had no role in study design, data collection and interpretation, or the decision to submit the work for publication.

### Author contributions

Yuji Nagai, Conceptualization, Formal analysis, Investigation, Visualization, Writing - original draft, Writing - review and editing; Yukiko Hori, Naohisa Miyakawa, Investigation, Writing - review and editing; Ken-ichi Inoue, Katsushi Kumata, Resources, Writing - review and editing; Toshiyuki Hirabayashi, Formal analysis, Investigation, Visualization, Writing - original draft, Writing - review and editing; Koki Mimura, Formal analysis, Writing - review and editing; Kei Oyama, Investigation, Visualization, Writing - review and editing; Yuki Hori, Haruhiko Iwaoki, Ming-Rong Zhang, Masahiko Takada, Makoto Higuchi, Writing - review and editing; Takafumi Minamimoto, Conceptualization, Writing - original draft, Project administration, Writing - review and editing

### Author ORCIDs

Yuji Nagai ⓘ https://orcid.org/0000-0001-7005-0749
Yukiko Hori ⓘ https://orcid.org/0000-0003-1023-9587
Takafumi Minamimoto ⓘ https://orcid.org/0000-0003-4305-0174

## Ethics

All experimental procedures involving animals were carried out in accordance with the Guide for the Care and Use of Nonhuman Primates in Neuroscience Research (The Japan Neuroscience Society; https://www.jnss.org/en/animal_primates) and were approved by the Animal Ethics Committee of the National Institutes for Quantum Science and Technology (Permit Number: 11-1038).

Reviewer #1 (Public review): https://doi.org/10.7554/eLife.105815.3.sa1
Reviewer #2 (Public review): https://doi.org/10.7554/eLife.105815.3.sa2
Reviewer #3 (Public review): https://doi.org/10.7554/eLife.105815.3.sa3
Author response https://doi.org/10.7554/eLife.105815.3.sa4

# Additional files

## Supplementary files

MDAR checklist

## Data availability

Source data to reproduce the main results of the paper presented in all figures are provided on GitHub (copy archived at *Nagai, 2025*).

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
