## [Editor Report · eLife Assessment]

This study provides novel and **fundamental** insights into the long-term use of DREADDs to modulate neuronal activity in nonhuman primates. The **exceptional** evidence demonstrates the peak dynamics and the subsequent stability of chemogenetic effects for 1.5 years, informing the experimental designs and the interpretation of highly impactful chemogenetic studies in macaques. The protocols, data, and outcomes can serve as guidelines for future experiments. Therefore, the findings will be of significant interest to the field of chemogenetics and may also be of broader interest to researchers and clinicians who seek to utilize viral vectors and/or related genetic technologies.

---

## [Referee Report · Reviewer #1 (Public review)]

Summary:

Inhibitory hM4Di and excitatory hM3Dq DREADDs are currently the most commonly utilized chemogenetic tools in the field of nonhuman primate research, but there is a lack of available information regarding the temporal aspects of virally-mediated DREADD expression and function. Nagai et al. investigated the longitudinal expression and efficacy of DREADDs to modulate neuronal activity in the macaque model. The authors demonstrate that both hM4Di and hM3Dq DREADDs reach peak expression levels after approximately 60 days and that stable expression was maintained for up to two years for hM4Di and at least one year for hM3Dq DREADDs. During this period, DREADDs effectively modulated neuronal activity, as evidenced by a variety of measures, including behavioural testing, functional imaging, and/or electrophysiological recording. Notably, some of the data suggest that DREADD expression may decline after two-three years. This is a novel finding and has important implications for the utilization of this technology for long-term studies, as well as its potential therapeutic applications. Lastly, the authors highlight that peak DREADD expression may be significantly influenced by the presence of fused or co-expressed protein tags, emphasizing the importance of careful design and selection of viral constructs for neuroscientific research. This study represents a critical step in the field of chemogenetics, setting the scene for future development and optimization of this technology.

Strengths:

The longitudinal approach of this study provides important preliminary insights into the long-term utility of chemogenetics, which has not yet been thoroughly explored.

The data presented are novel and inclusive, relying on well-established in vivo imaging methods as well as behavioral and immunohistochemical techniques. The conclusions made by the authors are generally supported by a combination of these techniques. In particular, the utilization of in vivo imaging as a non-invasive method is translationally relevant and likely to make an impact in the field of chemogenetics, such that other researchers may adopt this method of longitudinal assessment in their own experiments. Rigorous standards have been applied to the datasets, and the appropriate controls have been included where possible.

The number of macaque subjects (20) from which data was available is also notable. Behavioral testing was performed in 11 subjects, FDG-PET in 5, electrophysiology in 1, and [11C]DCZ-PET in 15. This is an impressive accumulation of work that will surely be appreciated by the growing community of researchers using chemogenetics in nonhuman primates.

The implication that chemogenetic effects can be maintained for up to 1.5-2 years, followed by a gradual decline beyond this period, is an important development in knowledge. The limited duration of DREADD expression may present an obstacle in the translation of chemogenetic technology as a potential therapeutic tool, and it will be of interest for researchers to explore whether this limitation can be overcome. This study therefore represents a key starting point upon which future research can build.

Weaknesses:

None.

---

## [Referee Report · Reviewer #2 (Public review)]

Summary:

This paper reports histological, PET imaging, functional and behavioural data evaluating the longevity of AAV2 infection in multiple brain areas of macaques in the context of DREADD experiments. The central aim is to provide unprecedented information about how long the expression of HM4di or HM3dq receptors are expressed and efficient in modulating brain functions after vector injections. The data show peak expression after 40 to 60 days of vector injection, and stable expressions for up to 1.5 years for hM4di, and that hM3dq remained mostly at 75% of peak after a year, declining to 50% after 2 years. DREADDs effectively modulated neuronal activity and behaviour for approximately two years, evaluated with behavioural testings, neural recordings or FDG-PET. A statistical evaluation revealed that vector titers, DREADD type and tags contribute to the measured peak level of DREADD expression.

The article present a thorough discussion of the limitations and specificities of chemogenetic approaches in monkeys.

Strength:

These are unique data, in non-human primate (NHP), an animal model that not only features physiological and immunological characteristics similar to humans, but also contributes to neurobiological functional studies over long timescales with experiments spanning months or years. This evaluation of long-term efficacy of DREADDs will be very important for all laboratories using chemogenetics in NHP but also for future use of such approach in experimental therapies. The longevity estimates are based on multiple approaches including behavioural and neurophysiological, thus providing information on functional efficacy of DREADD expression.

Performing such evaluation requires specific tools like PET imaging that very few monkey labs have access to. This study was done by the laboratory that has developed the radiotracer c11-DCZ, used here, a radiotracer binding selectively to DREADDs and providing, using PET, quantitative in vivo measures of DREADD expression. This study and its data should thus be a reference in the field, providing estimates to plan future chemogenetic experiments.

Publishing databases of experimental outcomes in NHP DREADD experiments is crucial for the community because such experiments are rare, expensive and long. It contributes to refining experiments and reducing the number of animals overall used in the domain.

Weaknesses:

This study is a meta-analysis of several experiments performed in one lab. The good side is that it combined a large amount of data that might not have been published individually; the down side is that all things where not planned and equated, creating a lot of unexplained variances in the data. However, this was judiciously used by the authors to provide very relevant information. One might think that organized multi-centric experiments planned using the knowledge acquired here, will provide help testing more parameters, including some related to inter-individual variability, and particular genetic constructs.

---

## [Referee Report · Reviewer #3 (Public review)]

Summary

This manuscript, from the developers of the novel DREADD-selective agonist DCZ (Nagai et al., 2020), utilizes a unique dataset where multiple PET scans in a large number of monkeys, including baseline scans before AAV injection, 30-120 days post-injection, and then periodically over the course of the prolonged experiments, were performed to access short- and long-term dynamics of DREADD expression in vivo, and to associate DREADD expression with the efficacy of manipulating the neuronal activity or behavior. The goal was to provide critical insights into practicality and design of multi-year studies using chemogenetics, and to elucidate factors affecting expression stability.

Strengths are systematic quantitative assessment of the effects of both excitatory and inhibitory DREADDs, quantification of both the short-term and longer-term dynamics, a wide range of functional assessment approaches (behavior, electrophysiology, imaging), and assessment of factors affecting DREADD expression levels, such as serotype, promoter, titer (concentration), tag, and DREADD type.

These finding will undoubtedly have a very significant impact on the rapidly growing, but still highly challenging field of primate chemogenetic manipulations. As such, the work represents an invaluable resource for the community.

---

## [Author Response]

The following is the authors’ response to the original reviews.

**Reviewer #1 (Public review):**
Overall, the conclusions of the paper are mostly supported by the data but may be overstated in some cases, and some details are also missing or not easily recognizable within the figures. The provision of additional information and analyses would be valuable to the reader and may even benefit the authors' interpretation of the data.

We thank the reviewer for the thoughtful and constructive feedback. We are pleased that the reviewer found the overall conclusions of our paper to be well supported by the data, and we appreciate the suggestions for improving figure clarity and interpretive accuracy. Below, we address each point with corresponding revisions.

The conclusion that DREADD expression gradually decreases after 1.5-2 years is only based on a select few of the subjects assessed; in Figure 2, it appears that only 3 hM4Di cases and 2 hM3Dq cases are assessed after the 2-year timepoint. The observed decline appears consistent within the hM4Di cases, but not for the hM3Dq cases (see Figure 2C: the AAV2.1-hSyn-hM3Dq-IRES-AcGFP line is increasing after 2 years.)

We agree that our interpretation should be stated more cautiously, given the limited number of cases assessed beyond the two-year timepoint. In the revised manuscript, we have clarified in the Results that the observed decline is based on a subset of animals. We have also included a text stating that while a consistent decline was observed in hM4Di-expressing monkeys, the trajectory for hM3Dq expression was more variable with at least one case showing an increased signal beyond two years.

Revised Results section:

Lines 140, “hM4Di expression levels remained stable at peak levels for approximately 1.5 years, followed by a gradual decline observed in one case after 2.5 years, and after approximately 3 years in the other two cases (Figure 2B, a and e/d, respectively). Compared with hM4Di expression, hM3Dq expression exhibited greater post-peak fluctuations. Nevertheless, it remained at ~70% of peak levels after about 1 year. This post-peak fluctuation was not significantly associated with the cumulative number of DREADD agonist injections (repeated-measures two-way ANOVA, main effect of activation times, F_(1,6)_ = 5.745, P = 0.054). Beyond 2 years post-injection, expression declined to ~50% in one case, whereas another case showed an apparent increase (Figure 2C, c and m, respectively).”

Given that individual differences may affect expression levels, it would be helpful to see additional labels on the graphs (or in the legends) indicating which subject and which region are being represented for each line and/or data point in Figure 1C, 2B, 2C, 5A, and 5B. Alternatively, for Figures 5A and B, an accompanying table listing this information would be sufficient.

We thank the reviewer for these helpful suggestions. In response, we have revised the relevant figures (Fig. 1C, 2B, 2C, and 5) as noted in the “Recommendations for the authors”, including simplifying visual encodings and improving labeling. We have also updated Table 2 to explicitly indicate the animal ID and brain regions associated with each data point shown in the figures.

While the authors comment on several factors that may influence peak expression levels, including serotype, promoter, titer, tag, and DREADD type, they do not comment on the volume of injection. The range in volume used per region in this study is between 2 and 54 microliters, with larger volumes typically (but not always) being used for cortical regions like the OFC and dlPFC, and smaller volumes for subcortical regions like the amygdala and putamen. This may weaken the claim that there is no significant relationship between peak expression level and brain region, as volume may be considered a confounding variable. Additionally, because of the possibility that larger volumes of viral vectors may be more likely to induce an immune response, which the authors suggest as a potential influence on transgene expression, not including volume as a factor of interest seems to be an oversight.

We thank the reviewer for raising this important issue. We agree that injection volume could act as a confounding variable, particularly since larger volumes were used in only handheld cortical injections. This overlap makes it difficult to disentangle the effect of volume from those of brain region or injection method. Moreover, data points associated with these larger volumes also deviated when volume was included in the model.

To address this, we performed a separate analysis restricted to injections delivered via microinjector, where a comparable volume range was used across cases. In this subset, we included injection volume as additional factor in the model and found that volume did not significantly impact peak expression levels. Instead, the presence of co-expressed protein tags remained a significant predictor, while viral titer no longer showed a significant effect. These updated results have replaced the originals in the revised Results section and in the new Figure 5. We have also revised the Discussion to reflect these updated findings.

The authors conclude that vectors encoding co-expressed protein tags (such as HA) led to reduced peak expression levels, relative to vectors with an IRES-GFP sequence or with no such element at all. While interesting, this finding does not necessarily seem relevant for the efficacy of long-term expression and function, given that the authors show in Figures 1 and 2 that peak expression (as indicated by a change in binding potential relative to non-displaced radioligand, or ΔBPND) appears to taper off in all or most of the constructs assessed. The authors should take care to point out that the decline in peak expression should not be confused with the decline in longitudinal expression, as this is not clear in the discussion; i.e. the subheading, "Factors influencing DREADD expression," might be better written as, "Factors influencing peak DREADD expression," and subsequent wording in this section should specify that these particular data concern peak expression only.

We appreciate this important clarification. In response, we have revised the title to "Protein tags reduce peak DREADD expression levels" in the Results section and “Factors influencing peak DREADD expression levels” in the Discussion section. Additionally, we specified that our analysis focused on peak ΔBP_ND_ values around 60 days post-injection. We have also explicitly distinguished these findings from the later-stage changes in expression seen in the longitudinal PET data in both the Results and Discussion sections.

**Reviewer #1 (Recommendations for the authors):**
(1) Will any of these datasets be made available to other researchers upon request?

All data used to generate the figures have been made publicly available via our GitHub repository (https://github.com/minamimoto-lab/2024-Nagai-LongitudinalPET). This has been stated in the "Data availability" section in the revised manuscript.

(2) Suggested modifications to figures:a) In Figures 2B and C, the inclusion of "serotype" as a separate legend with individual shapes seems superfluous, as the serotype is also listed as part of the colour-coded vector

We agree that the serotype legend was redundant since this information is already included in the color-coded vector labels. In response, we have removed the serotype shape indicators and now represent the data using only vector-construct-based color coding for clarity in Figure 2B and C.

b) In Figures 3A and B, it would be nice to see tics (representing agonist administration) for all subjects, not just the two that are exemplified in panels C-D and F-H. Perhaps grey tics for the non-exemplified subjects could be used.

In response, we have included black and white ticks to indicate all agonist administration across all subjects in Figure 3A and B, with the type of agonist clearly specified.

c) In Figure 4C, a Nissl- stained section is said to demonstrate the absence of neuronal loss at the vector injection sites. However, if the neuronal loss is subtle or widespread, this might not be easily visualized by Nissl. I would suggest including an additional image from the same section, in a non-injected cortical area, to show there is no significant difference between the injected and non-injected region.

To better demonstrate the absence of neuronal loss at the injection site, we have included an image from the contralateral, non-injected region of the same section for comparison (Fig. 4C).

d) In Figure 5A: is it possible that the hM3Dq construct with a titer of 5×10^13 gc/ml is an outlier, relative to the other hM3Dq constructs used?

We thank the reviewer for raising this important observation. To evaluate whether the high-titer constructs represented a statistical outlier that might artifactually influence the observed trends, we performed a permutation-based outlier analysis. This assessment identified this point in question, as well as one additional case (titer 4.6 x 10e13 gc/ml, #255, L_Put), as significant outlier relative to the distribution of the dataset.

Accordingly, we excluded these two data points from the analysis. Importantly, this exclusion did not meaningfully alter the overall trend or the statistical conclusions—specifically, the significant effect of co-expressed protein tags on peak expression levels remain robust. We have updated the Methods section to describe this outlier handling and added a corresponding note in the figure legend.

**Reviewer #2 (Public review):**
WeaknessesThis study is a meta-analysis of several experiments performed in one lab. The good side is that it combined a large amount of data that might not have been published individually; the downside is that all things were not planned and equated, creating a lot of unexplained variances in the data. This was yet judiciously used by the authors, but one might think that planned and organized multicentric experiments would provide more information and help test more parameters, including some related to inter-individual variability, and particular genetic constructs.

We thank the reviewer for bringing this important point to our attention. We fully acknowledge that the retrospective nature of our dataset—compiled from multiple studies conducted within a single laboratory—introduces variability related to differences in injection parameters and scanning timelines. While this reflects the practical realities and constraints of long-term NHP research, we agree that more standardized and prospectively designed studies would better control such source of variances. To address this, we have added the following statement to the "Technical consideration" section in Discussion:

Lines 297, "This study included a retrospective analysis of datasets pooled from multiple studies conducted within a single laboratory, which inherently introduced variability across injection parameters and scan intervals. While such an approach reflects real-world practices in long-term NHP research, future studies, including multicenter efforts using harmonized protocols, will be valuable for systematically assessing inter-individual differences and optimizing key experimental parameters."

**Reviewer #2 (Recommendations for the authors):**
I just have a few minor points that might help improve the paper:(1) Figure 1C y-axis label: should add deltaBPnd in parentheses for clarity.

We have added “ΔBP_ND_” to the y-axis label for clarity.

The choice of a sigmoid curve is the simplest clear fit, but it doesn't really consider the presence of the peak described in the paper. Would there be a way to fit the dynamic including fitting the peak?

We agree that using a simple sigmoid curve for modeling expression dynamics is a limitation. In response to this and a similar comment from Reviewer #3, we tested a double logistic function (as suggested) to see if it better represented the rise and decline pattern. However, as described below, the original simple sigmoid curve was a better fit for the data. We have included a discussion regarding this limitation of this analysis. See Reviewer #3 recommendations (2) for details.

The colour scheme in Figure 1C should be changed to make things clearer, and maybe use another dimension (like dotted lines) to separate hM4Di from hM3Dq.

We have improved the visual clarity of Figure 1C by modifying the color scheme to represent vector construct and using distinct line types (dashed for hM4Di and solid for hM3Dq data) to separate DREADD type.

(2) Figure 2I don't understand how the referencing to 100 was made: was it by selecting the overall peak value or the peak value observed between 40 and 80 days? If the former then I can't see how some values are higher than the peak. If the second then it means some peak values occurred after 80 days and data are not completely re-aligned.

We thank the reviewer for the opportunity to clarify this point. The normalization was based on the peak value observed between 40–80 days post-injection, as this window typically captured the peak expression phase in our dataset (see Figure 1). However, in some long-term cases where PET scans were limited during this period—e.g., with one scan performing at day 40—it is possible that the actual peak occurred later. Therefore, instances where ΔBP_ND_ values slightly exceeded the reference peak at later time points likely reflect this sampling limitation. We have clarified this methodological detail in the revised Results section to improve transparency.

The methods section mentions the use of CNO but this is not in the main paper which seems to state that only DCZ was used: the authors should clarify this

Although DCZ was the primary agonist used, CNO and C21 were also used in a few animals (e.g., monkeys #153, #221, and #207) for behavioral assessments. We have clarified this in the Results section and revised Figure 3 to indicate the specific agonist used for each subject. Additionally, we have updated the Methods section to clearly specify the use and dosage of DCZ, CNO, and C21, to avoid any confusion regarding the experimental design.

**Reviewer #3 (Public review):**

Minor weaknesses are related to a few instances of suboptimal phrasing, and some room for improvement in time course visualization and quantification. These would be easily addressed in a revision.

These findings will undoubtedly have a very significant impact on the rapidly growing but still highly challenging field of primate chemogenetic manipulations. As such, the work represents an invaluable resource for the community.

We thank the reviewer for the positive assessment of our manuscript and for the constructive suggestions. We address each comment in the following point-by-point responses and have revised the manuscript accordingly.

**Reviewer #3 (Recommendations for the authors):**
(1) Please clarify the reasoning was, behind restricting the analysis in Figure 1 only to 7 monkeys with subcortical AAV injection?

We focused the analysis shown in Figure 1 on 7 monkeys with subcortical AAV injections who received comparative injection volumes. These data were primary part of vector test studies, allowing for repeated PET scans within 150 days post-injection. In contrast, monkeys with cortical injections—including larger volumes—were allocated to behavioral studies and therefore were not scanned as frequently during the early phase. We will clarify this rationale in the Results section.

(2) Figure 1: Not sure if a simple sigmoid is the best model for these, mostly peaking and then descending somewhat, curves. I suggest testing a more complex model, for instance, double logistic function of a type f(t) = a + b/(1+exp(-c*(t-d))) - e/(1+exp(-g*(t-h))), with the first logistic term modeling the rise to peak, and the second term for partial decline and stabilization

We appreciate the reviewer’s thoughtful suggestion to use a double logistic function to better model both the rising and declining phases of the expression curve. In response to this and similar comments from Reviewer #1, we tested the proposed model and found that, while it could capture the peak and subsequent decline, the resulting fit appeared less biologically plausible (See below). Moreover, model comparison using BIC favored the original simple sigmoid model (BIC = 61.1 vs. 62.9 for the simple and double logistic model, respectively). This information has been included in the revised figure legend for clarity.

Given these results, we retained the original simple sigmoid function in the revised manuscript, as it provides a sufficient and interpretable approximation of the early expression trajectory—particularly the peak expression-time estimation, which was the main purpose of this analysis. We have updated the Methods section to clarify our modeling and rationale as follows:

Lines 530, "To model the time course of DREADD expression, we used a single sigmoid function, referencing past in vivo fluorescent measurements (Diester et al., 2011). Curve fitting was performed using least squares minimization. For comparison, a double logistic function was also tested and evaluated using the Bayesian Information Criterion (BIC) to assess model fit."

We also acknowledge that a more detailed understanding of post-peak expression changes will require additional PET measurements, particularly between 60- and 120-days post-injection, across a larger number of animals. We have included this point in the revised Discussion to highlight the need for future work focused on finer-grained modeling of expression decline:

Lines 317, “Although we modeled the time course of DREADD expression using a single sigmoid function, PET data from several monkeys showed a modest decline following the peak. While the sigmoid model captured the early-phase dynamics and offered a reliable estimate of peak timing, additional PET scans—particularly between 60- and 120-days post-injection—will be essential to fully characterize the biological basis of the post-peak expression trajectories.”

**Author response image 1. sa4fig1:** 

(3) Figure 2: It seems that the individual curves are for different monkeys, I counted 7 in B and 8 in C, why "across 11 monkeys"? Were there several monkeys both with hM4Diand hM3Dq? Does not look like that from Table 1. Generally, I would suggest associating specific animals from Tables 1 and 2 to the panels in Figures 1 and 2.

Some animals received multiple vector types, leading to more curves than individual subjects. We have revised the figure legends and updated Table 2 to explicitly relate each curve with the specific animal and brain region.

(4) I also propose plotting the average of (interpolated) curves across animals, to convey the main message of the figure more effectively.

We agree that plotting the mean of the interpolated expression curves would help convey the group trend. We added averaged curves to Figure 2BC.

(5) Similarly, in line 155 "We assessed data from 17 monkeys to evaluate ... Monkeys expressing hM4Di were assessed through behavioral testing (N = 11) and alterations in neuronal activity using electrophysiology (N = 2)..." - please explain how 17 is derived from 11, 2, 5 and 1. It is possible to glean from Table 1 that it is the calculation is 11 (including 2 with ephys) + 5 + 1 = 17, but it might appear as a mistake if one does not go deep into Table 1.

We have clarified in both the text and Table 1 that some monkeys (e.g., #201 and #207) underwent both behavioral and electrophysiological assessments, resulting in the overlapping counts. Specifically, the dataset includes 11 monkeys for hM4Di-related behavior testing (two of which underwent electrophysiology testing), 5 monkeys assessed for hM3Dq with FDG-PET, and 1 monkey assessed for hM3Dq with electrophysiology, totaling 19 assessments across 17 monkeys. We have revised the Results section to make this distinction more explicit to avoid confusion, as follows:

Lines 164, "Monkeys expressing hM4Di (N = 11) were assessed through behavioral testing, two of which also underwent electrophysiological assessment. Monkeys expressing hM3Dq (N = 6) were assessed for changes in glucose metabolism via [^18^F]FDG-PET (N = 5) or alterations in neuronal activity using electrophysiology (N = 1).”

(6) Line 473: "These stock solutions were then diluted in saline to a final volume of 0.1 ml (2.5% DMSO in saline), achieving a dose of 0.1 ml/kg and 3 mg/kg for DCZ and CNO, respectively." Please clarify: the injection volume was always 0.1 ml? then it is not clear how the dose can be 0.1 ml/kg (for a several kg monkey), and why DCZ and CNO doses are described in ml/kg vs mg/kg?

We thank the reviewer for pointing out this ambiguity. We apologize for the oversight and also acknowledge that we omitted mention of C21, which was used in a small number of cases. To address this, we have revised the “Administration of DREADD agonist” section of the Methods to clearly describe the preparation, the volume, and dosage for each agonist (DCZ, CNO, and C21) as follows:

Lines 493, “Deschloroclozapine (DCZ; HY-42110, MedChemExpress) was the primary agonist used. DCZ was first dissolved in dimethyl sulfoxide (DMSO; FUJIFILM Wako Pure Chemical Corp.) and then diluted in saline to a final volume of 1 mL, with the final DMSO concentration adjusted to 2.5% or less. DCZ was administered intramuscularly at a dose of 0.1 mg/kg for hM4Di activation, and at 1–3 µg/kg for hM3Dq activation. For behavioral testing, DCZ was injected approximately 15 min before the start of the experiment unless otherwise noted. Fresh DCZ solutions were prepared daily.

In a limited number of cases, clozapine-N-oxide (CNO; Toronto Research Chemicals) or Compound 21 (C21; Tocris) was used as an alternative DREADD agonist for some hM4Di experiments. Both compounds were dissolved in DMSO and then diluted in saline to a final volume of 2–3 mL, also maintaining DMSO concentrations below 2.5%. CNO and C21 were administered intravenously at doses of 3 mg/kg and 0.3 mg/kg, respectively.”

(7) Figure 5A: What do regression lines represent? Do they show a simple linear regression (then please report statistics such as R-squared and p-values), or is it related to the linear model described in Table 3 (but then I am not sure how separate DREADDs can be plotted if they are one of the factors)?

We thank the reviewer for the insightful question. In the original version of Figure 5A, the regression lines represented simple linear fits used to illustrate the relationship between viral titer and peak expression levels, based on our initial analysis in which titer appeared to have a significant effect without any notable interaction with other factors (such as DREADD type).

However, after conducting a more detailed analysis that incorporated injection volume as an additional factor and excluded cortical injections and statistical outliers (as suggested by Reviewer #1), viral titer was no longer found to significantly predict peak expression levels. Consequently, we revised the figure to focus on the effect of reporter tag, which remained the most consistent and robust predictor in our model.

In the updated Figure 5, we have removed the relationship between viral titer and expression level with regression lines.